# Safety of Fingolimod in Patients with Multiple Sclerosis Switched from Natalizumab: Results from TRANSITION―A 2-Year, Multicenter, Observational, Cohort Study

**DOI:** 10.3390/brainsci12020215

**Published:** 2022-02-04

**Authors:** Helmut Butzkueven, Paul S. Giacomini, Stanley Cohan, Tjalf Ziemssen, Daniel Sienkiewicz, Ying Zhang, Yvonne Geissbühler, Diego Silva, Davorka Tomic, Harald Kropshofer, Maria Trojano

**Affiliations:** 1Department of Neuroscience, Monash University, Melbourne, VIC 3004, Australia; 2Department of Neurology, Alfred Hospital, Melbourne, VIC 3004, Australia; 3Department of Neurology and Neurosurgery, McGill University, Montreal, QC H3A 0G4, Canada; paul.giacomini@mcgill.ca; 4Multiple Sclerosis Clinic, Montreal Neurological Institute and Hospital, Montreal, QC H3A 2B4, Canada; 5Providence Multiple Sclerosis Center, Portland, OR 97225, USA; stanley.cohan@providence.org; 6Center of Clinical Neurosciences, University Hospital Carl Gustav Carus, 01307 Dresden, Germany; tjalf.ziemssen@uniklinikum-dresden.de; 7Novartis Pharmaceuticals Corporation, East Hanover, NJ 07936, USA; daniel.sienkiewicz@novartis.com (D.S.); ying4.zhang@novartis.com (Y.Z.); 8Novartis Pharma AG, 4056 Basel, Switzerland; yvonne.geissbuehler@novartis.com (Y.G.); diesilva@hotmail.com (D.S.); davorka@gmx.ch (D.T.); harald.kropshofer@novartis.com (H.K.); 9Department of Basic Medical Sciences, Neurosciences and Sense Organs, University of Bari, 70121 Bari, Italy; maria.troiano@uniba.it

**Keywords:** fingolimod, natalizumab, switching, transition, observational study

## Abstract

Multiple sclerosis (MS) patients receiving natalizumab and who are at risk of developing progressive multifocal leukoencephalopathy (PML) often switch to other high-efficacy disease-modifying therapies including fingolimod as a risk mitigation strategy, which could impact treatment safety and effectiveness. The TRANSITION study aimed to evaluate the safety of fingolimod over two years in patients with MS after switching from natalizumab in a real-world setting. The safety and effectiveness were assessed by monitoring serious and other adverse events (SAEs, AEs). We assessed effectiveness by recording relapses, Expanded Disability Status Scale (EDSS) scores, and MRI activity. Of 637 patients enrolled, 505 completed the study (mean age, 42 years). Overall, 72.8% and 12.7% experienced AEs and SAEs respectively. The most common AEs were fatigue, headache, and urinary tract infection; no cases of PML were observed. Fingolimod treatment resulted in low disease activity. Patients with ≤8 weeks washout period had a markedly lower risk of relapses (4.5%) than those with >8 weeks (51.4%). In patients switching from natalizumab to fingolimod, no new safety signals with overall low relapse activity were observed in patients with washout latencies of ≤8 weeks before fingolimod initiation. Fingolimod was found to be safe and effective in patients transitioning from natalizumab.

## 1. Introduction

Fingolimod, a sphingosine 1-phosphate receptor modulator, is approved as a once daily oral therapy for relapsing forms of MS, with proven efficacy and safety in clinical trials and real-world settings [1,2,3,4,5]. The efficacy and safety of fingolimod have been reported in treatment-naïve patients as well as in patients switching to fingolimod from interferon (IFN) β-1a and glatiramer acetate [6,7].

Natalizumab, a selective adhesion-molecule inhibitor, is approved for the treatment of relapsing-remitting MS as infusion therapy every four weeks. In Phase 3 clinical trials, natalizumab significantly reduced relapses and lesions compared with placebo and IFN β-1a [8,9]. However, treatment with natalizumab is associated with a significant risk of progressive multifocal leukoencephalopathy (PML), an opportunistic infection caused by the reactivation of latent John Cunningham virus (JCV) [10]. Patients receiving natalizumab treatment for more than two years who have a history of prior immunosuppressant use, or are seropositive for anti-JCV antibodies, are at increased risk of PML [10]. These patients often switch to other high-efficacy disease-modifying therapies (DMTs), including fingolimod, as a risk mitigation strategy [11].

As of 28 February 2021, 45 PML cases associated with fingolimod treatment have been identified in the post-marketing setting, with an overall incidence rate of approximately 1.4:10,000 patients [12]. The safety in patients switching from natalizumab (mean half-life: 11 ± 4 days) to fingolimod is an open question due to the concern that natalizumab and fingolimod may potentiate their respective immunosuppressive effects in these patients. In clinical practice, this is counterbalanced by the concern that long wash-out periods during medication switching are known to increase the risk of relapses [13]. The present study evaluated the overall safety profile of fingolimod treatment over a period of two years in patients who were previously treated with natalizumab.

## 2. Methods

### 2.1. Study Design

TRANSITION was a global, multicenter, 2-year, prospective, observational, single-cohort study in patients with relapsing MS, previously treated with natalizumab, who switched to fingolimod in routine clinical practice (Appendix A). This observational study was designed and implemented following the Good Pharmacoepidemiology Practice guidelines for observational studies of the International Society of Pharmacoepidemiology (IPSE 2008) [14], the STROBE (Strengthening the Reporting of Observational Studies in Epidemiology) guidelines [15], and with the ethical principles laid down by the Declaration of Helsinki. Written informed consent was obtained from all patients.

### 2.2. Patient Population

Patients with relapsing MS were eligible for enrolment if they had received their last infusion of natalizumab within the last 12 months and had either initiated fingolimod treatment within the last 12 months or were about to commence fingolimod at the beginning of the study. Fingolimod was prescribed in compliance with the local prescribing information and routine medical care. Patients who were previously treated with fingolimod at any time other than the 12 months prior to the study entry were excluded.

### 2.3. Study Assessments

The main objective of the study was to explore the overall safety profile of fingolimod treatment sequenced after natalizumab, assessed by adverse events (AEs and SAEs) and vital signs. AEs were observed from the date of first administration of fingolimod after switching from natalizumab, even if that occurred prior to study entry. In addition, the incidence of serious adverse events (SAEs) and AEs leading to treatment discontinuation were reported.

### 2.4. Other Assessments

Other safety events of special interest included serious infections (opportunistic infections such as PML), cardiac and vascular events (e.g., stroke, myocardial infarction, angina pectoris and peripheral vascular disease, second and third degree atrioventricular [AV] block, hypertension, symptomatic bradyarrhythmias), macular edema, liver enzyme levels, pulmonary events, malignancies, seizures, atypical MS relapses, and atypical severe neurological events. In addition, first-dose monitoring data and clinically significant laboratory abnormalities were reported.

The effectiveness of fingolimod treatment on MS disease activity was evaluated by relapse incidence, Expanded Disability Status Scale (EDSS) scores and MRI activity (number of gadolinium-enhancing [Gd+] T1 lesions and new or enlarging T2 lesions) over 2 years.

Health-related quality of life outcome measures included the Multiple Sclerosis Impact Scale (MSIS-29, assessed at baseline, Months 6 and 12, and at the end of study), the Treatment Satisfaction Questionnaire for Medication-9 (TSQM-9, at baseline and Month 6), and the Medication Preference Questionnaire at Month 6. Authorization for use of these questionnaires was received. Patient retention and discontinuations from fingolimod treatment were monitored throughout the study period.

### 2.5. Sample Size and Statistical Analysis

Given the background incidence rate for serious infections in patients receiving placebo of approximately 9 per 1000 patient-years (PY), a sample size of 500 has an 80% or 90% power to detect a relative risk of 2.4 or 2.6 above the background rate, respectively. Patient disposition, demographics, and baseline characteristics were summarized for all patients who provided informed consent and enrolled in the study. We included all patients who had received at least one dose of fingolimod in the safety and effectiveness analyses. To further support the study findings, we carried out a post-hoc analysis separating patients who initiated fingolimod at the time of study entry (prospective group) or prior to study entry (retrospective group).

Descriptive statistics were used to summarize safety variables. Incidence rates (IRs) and exposure-adjusted IRs per 100 PY of AEs, SAEs, and AEs leading to permanent discontinuation were summarized by primary system organ class and preferred term (PT). For AEs and SAEs within subgroups, occurrence rate ratios were calculated using negative binomial regression model with log (duration of fingolimod exposure in years) as offset variable. The model was adjusted for natalizumab washout period, natalizumab exposure, prior use of immunosuppressive agents, and JCV status. Relapses, EDSS scores, and MRI data were summarized using descriptive statistics; group-and patient-level ARRs were analyzed in all patients. Further details are provided in Appendix A. Negative binomial regression was used to estimate the ARRs, adjusted for the number of relapses experienced in the last 2 years before first fingolimod dose, natalizumab washout period, and baseline EDSS. ARRs were also analyzed by duration of fingolimod exposure and duration of washout period.

Safety and effectiveness data were analyzed in the following subgroups: duration of washout period between natalizumab and fingolimod initiation (≤8 weeks and >8 weeks); duration of exposure to natalizumab (<2 years and ≥2 years); prior use of immunosuppressive agents (Yes or No); and JCV status (positive or negative). Two independent Pearson’s Chi-square tests were used to assess the relationship between washout period, JCV status, and use of immunosuppressive agents before the study entry.

## 3. Results

### 3.1. Patient Disposition and Baseline Characteristics

Of 637 patients enrolled (first patient, first visit: 25 September 2012; last patient last visit: 31 January 2017), 361 were in the prospective group and 266 were enrolled retrospectively (details of the enrollment centers are provided in Appendix A). A total of 505 patients completed the study, and 132 patients prematurely discontinued (Figure 1). At baseline, the mean age of the study population was 42.0 years and 71.9% were women (Table 1). The median exposure to natalizumab was 924.0 days (or 2.53 years), range 1–3071 days. The primary reasons for switching from natalizumab to fingolimod were JCV-positive serology (68.3%) and >2 years of natalizumab treatment (30.3%). In general, baseline demographics and disease characteristics were comparable between the prospective and retrospective groups. Further details are provided in the Appendix A.

### 3.2. Safety Outcomes

Fingolimod exposure was calculated from the first dose of fingolimod, even if it was initiated prior to study entry. The median (range) duration of fingolimod exposure was 697.5 (1–1103) days with an overall exposure of 1001 PY. The majority of patients (86.8%) were exposed to fingolimod for ≥180 days and three quarters of enrolled patients received fingolimod treatment for ≥360 days.

Overall, 457 patients (72.8% total; 75.4% prospective group, 69.6% retrospective group) reported ≥1 AE and the corresponding overall IR (95% CI) of AEs during the study was 91.5 (83.3; 100.3) per 100 patient years (Table 2). The incidence rate of overall AEs was higher in the prospective group than in the retrospective group. The most common AEs by preferred term (PT) were fatigue, headache, and urinary tract infection. The exposure-adjusted IR (95% CI) of AEs during fingolimod treatment was 172.3/100 patient years for ≤180 days, 59.5 for >180 to ≤360 days, and 30.1 for >360 days drug exposure.

A total of 80 patients (12.7%) experienced ≥1 SAE, and the overall IR (95% CI) was 6.9 (5.5; 8.6; Table 2). The most commonly reported SAE by PT was MS relapse (2.1%). Three deaths occurred during the study and the reasons were septic shock (secondary to pneumonia; only this event was considered related to fingolimod treatment), ventricular fibrillation, and aspiration pneumonia. Time to onset of death ranged from approximately 1 year to 2.3 years. A total of 58 patients (9.2%) experienced ≥1 AE that led to treatment discontinuation (overall IR (95% CI): 4.9 (3.7; 6.3)) and the most common (≥5 patients) reasons were lymphopenia and fatigue (five patients each; 0.80%, IR: 0.40 per 100 PY).

#### 3.2.1. Vital Signs

There were no clinically relevant changes in vital signs in fingolimod-treated patients over two years. Approximately 21% of patients had an increase (≥15 bpm) and 15% had a decrease (<15 bpm) in sitting pulse rate, 27% had an increase (≥20 mm Hg) and 13% had a decrease (<20 mm Hg) in sitting blood pressure, and 24% had an increase (≥15 mm Hg) and 15% had a decrease (<15 mm Hg) in diastolic blood pressure.

#### 3.2.2. Selected Safety Outcomes

The most frequently reported selected safety events of special interest (cut-off IR > 5) were MS relapse (31.4%; IR, 18.36), cardiovascular AEs (23.3%; 14.69), asthenia (12.6%; 7.33), upper respiratory tract infections (11.8%; 6.78), and urinary tract infections (10.4%; 5.85). No cases of PML occurred during the course of the study.

The most frequently reported serious selected events of special interest (cut-off IR > 1) were infections (3.3%; IR, 1.8), MS relapse (2.7%; 1.46), and atypical MS relapse (2.6%; 1.37). Unfortunately, information beyond the label “atypical relapse” for these events was not collected in the CRF.

#### 3.2.3. First-Dose Observation

First-dose monitoring data were collected prospectively from patients who started fingolimod treatment at the start of the study and retrospectively from those who initiated fingolimod prior to study entry. A decrease in pulse rate was observed, with the lowest pulse rate between 3 and 5 h. Six cases of clinically significant electrocardiogram (ECG) abnormalities were observed, all of which were asymptomatic and all patients recovered. Bradyarrhythmia was reported in 53 patients (8.4%) and the majority of them (34 patients) had this AE after Day 7 of fingolimod treatment initiation. These include first degree AV block and sinus bradycardia in three patients each and second degree AV block and dizziness in two patients each on Day 1 of fingolimod initiation. No high degree AV blocks or Mobitz II events were observed.

#### 3.2.4. Laboratory Abnormalities

After fingolimod initiation, approximately 10% (18/182) of patients had aspartate aminotransferase or alanine aminotransferase (ALT) levels between >3 × upper limit of normal (ULN) and ≤5 × ULN, 1.6% (3/182) had levels between >5 and ≤8 × ULN, 1.1% (2/182) had levels between >10 and ≤20 × ULN, and 0.5% (1/182) had >20 × ULN. None of the patients met the Hy’s law criteria [16].

#### 3.2.5. Safety Outcomes in Subgroups

There was no difference in the occurrence of AEs between different subgroups (washout period of ≤8 weeks and >8 weeks, duration of exposure to natalizumab <2 years and ≥2 years and JCV status positive or negative) except that patients with no prior exposure to immunosuppressive agents reported higher overall occurrence of AEs compared to those with prior exposure. Similarly, there was no difference in the occurrence of SAEs between these subgroups. The occurrence rate ratios (95% CIs) for AEs and SAEs in all of the subgroups (Appendix A) and most frequent AEs and SAEs (Appendix A) are provided in the Appendix A. Further, regardless of JCV status (positive or negative), there was no association of SAE rate with washout period (*p* = 0.608) and use of an immunosuppressive agent before the study entry (*p* = 0.136).

### 3.3. Effectiveness Outcomes

In total, 202 (32.2%) patients experienced 332 relapses during the study, which included the washout period, while most patients (67.8%) were relapse-free until the end of the study. Most of the relapses were treated (287 relapses, 86.4%) and recovered either completely (163 relapses, 49.1%) or partially (89 relapses, 26.8%). The mean (standard deviation (SD)) duration of the relapse was 32.9 (30.93) days. The post hoc analysis observed 188 relapses in 117 patients in the prospective group and 144 relapses in 85 patients in the retrospective group. Further details are provided in the Appendix A.

Before fingolimod initiation, during the washout period, relapses were observed in 14.2% of patients. The proportion of patients with relapses increased with the length of washout period at all studied timepoints: 4.5% in ≤8 weeks, 7.8% in >8 weeks and ≤12 weeks, 11.7% in >12 weeks and ≤16 weeks, and 31.9% in >16 weeks (for all, *p* < 0.0001).

There was no difference in proportion of patients experiencing on-study relapses between subgroups of patients with a washout period of ≤8 weeks versus >8 weeks (36.5% vs. 30.9%, *p* = 0.188). Among the patients who had MS relapses, 20.5% of the patients in the ≤8 weeks group recovered completely and data were unknown/missing in 5.8% of patients; in the >8 weeks group, 17.2% of patients recovered completely and data were unknown/missing in 2.3% of patients.

After initiation of fingolimod, 326 relapses were reported during the study with a group-level ARR of 0.275 (prospective group, 0.27; retrospective group, 0.282) and a mean (SD) patient-level ARR of 0.33 ((0.82); prospective group, 0.32 (0.87); retrospective group, 0.34 (0.74)). After adjusting for the number of relapses experienced in the last two years before first fingolimod dose, natalizumab washout period and the baseline EDSS score, overall ARR was 0.244 (95% CI: 0.20; 0.30). When analyzed by subgroups, both the group-level (0.34 vs. 0.26) and patient-level (0.53 vs. 0.27) ARR was higher for patients with ≤8 weeks of washout period compared with the patients with >8 weeks of washout period.

Over time, the change in mean EDSS score from baseline increased by 0.14 at Month 12 and 0.15 at Month 24 in the overall population (Figure 2). Among subgroups of patients, no difference in EDSS change was observed in patients with ≤8 weeks compared with those with >8 weeks of washout period over 24 months of study period (*p* = NS at all timepoints).

Since the previous scan, worsening in the MRI scan was reported in 18.1% of patients at baseline. Similarly, 20.5% and 25.9% of patients experienced worsening at Months 12 and 24, respectively (Table 3). After fingolimod treatment, 71.4% of patients were free from Gd+ T1 lesions at Month 12 and 62.5% were free at Month 24. The number of T2 lesions increased in more than 50% of patients, at both Months 12 and 24.

### 3.4. Patient-Reported Outcomes

No notable change was observed in the overall MSIS-29 scale. The mean change in total MSIS-29 score from baseline of 35.9 to the end of the study was 0.5 (*p* = NS; Figure 3A). No significant change was observed in the physical and psychological impact scales from baseline to end of the study (*p* = NS).

Global satisfaction score, assessed by TSQM-9, slightly decreased from baseline to Month 6 (n = 160), whereas the effectiveness and convenience domains scores increased slightly from baseline to Month 6 (Figure 3B). Of the 167 patients that responded to the treatment preference question at Month 6, 47 (28.1%) preferred treatment with natalizumab while 120 (71.9%) preferred treatment with fingolimod.

## 4. Discussion

In this two-year observational study, the safety of fingolimod in patients switching from natalizumab were found to be consistent with previous findings from the clinical and real-world studies of fingolimod [1,2,3,4,5]. Fingolimod was found to be safe and effective in patients transitioning from the natalizumab treatment, with low disease activity.

Overall, 72.8% of patients experienced AEs in the study. The most commonly reported AEs were fatigue, urinary tract infection, lymphopenia, and headache. These findings were similar to those observed in TOFINGO study [17], and indicate that switching patients to fingolimod from natalizumab does not lead to higher incidence of AEs or SAEs than those observed in other fingolimod trials [1,2,3]. No new or unexpected safety signals were observed, thereby ruling out potential additive or overlapping effects on the immune system by fingolimod treatment in patients previously treated with natalizumab.

We chose to investigate the relationship of washout period to AE and SAE incidence because of the known long elimination half-life of natalizumab and reported prolonged effects of natalizumab on immune surveillance [10]. The median alpha4 integrin receptor occupancy after the last dose of natalizumab was approximately 80% at 8 weeks, which declined to 31% at 12 weeks [18]. Similarly in other study, receptors levels reached a plateau of 10% to 15% by Week 16 of last natalizumab dose [19].

In our study, there was no difference in the occurrence of AEs in patients with ≤8 weeks washout period compared with >8 weeks washout period. We could not replicate the results of the TOFINGO study, where a higher incidence of AEs was observed in the 8-week washout group during fingolimod therapy (compared to longer washout periods), mainly due to a higher incidence of relatively minor infections including nasopharyngitis and urinary tract infections [17]. The safety data from the other subgroups in this study further indicate that switching patients from natalizumab to fingolimod does not lead to a higher incidence of AEs or SAEs during the study regardless of their natalizumab treatment duration, prior exposure to immunosuppressive agents and JCV status.

We observed no cases of opportunistic infections, including PML. Our study had very low power to detect PML risk differences, as the overall PML risk associated with fingolimod not attributed to previous NTZ exposure is considerably lower (in the range of 1/10,000) than that observed with natalizumab in anti-JCV Ab positive patients [20].

In line with results from pivotal studies, asymptomatic elevations of liver enzymes were observed in <10% of patients, mainly ALT, that are reversible after fingolimod discontinuation [2,3]. No patients met the criteria for Hy’s law.

In our study, approximately two-thirds of patients were free from relapses until the end of the study. Patients with a shorter washout period (≤8 weeks) before initiating fingolimod had a lower risk of relapses than those with a longer washout period (>8 weeks). This replicates several prior studies [17,21,22,23], leading to the recommendation that fingolimod should be commenced within 4–8 weeks of natalizumab discontinuation [21,24,25,26,27]. We could not detect any safety signal related to <8 weeks washout periods. Thus, reducing the washout period to eight weeks or less outweighs the potential risk of infections, as previously described [22]. However, baseline cerebral MRI prior to fingolimod initiation to detect subclinical PML is recommended. Some authors even recommend testing of JCV DNA in the CSF of patients with anti-JCV Ab seropositivity, though this is controversial [24].

In the current study, fingolimod treatment maintained low ARR, and most patients (427/628) remained relapse-free until the end of the study. Similar results with a favorable benefit-risk profile over 48 months of fingolimod treatment were observed in the PANGAEA study [25]. EDSS scores followed a similar dynamic; slight increase of mean EDSS scores during the study could be related to breakthrough relapses with residual disability increase in patients with longer washout periods after natalizumab cessation.

A few studies have reported the recurrence of disease activity during the washout period or immediately after initiating fingolimod therapy [21,28]. However, very few studies evaluated the long-term treatment effect of fingolimod on disease activity after switching from natalizumab [25]. These results suggest that fingolimod is effective in controlling disease activity in the long term.

In the current study, we observed that a higher proportion of patients (71.9%) preferred fingolimod over natalizumab at six months. These results are consistent with results from the TOFINGO study, where >80% of patients preferred fingolimod over natalizumab (17). The TOFINGO study assessed safety outcomes for 24 weeks of fingolimod exposure while our study evaluated the safety exposure for longer period of 24 months [17].

Our study was limited by the nature of its study design–an open-label treatment, single-arm study without a comparator arm. Another limiting factor may be the discontinuation rate of 20.7% over a period of 24 months. The interpretation of study results in overall population may be confounded by the data from the retrospective patients who initiated the fingolimod before study start due to observational design. Safety results in the retrospective group may also be affected by their correct diagnoses and may not account for safety events at the time of treatment initiation. However, serious adverse events and treated relapses are not likely to be under-reported in the retrospective group, due to their routine documentation in the medical record. We further examined this by carrying out a post hoc analysis separating the prospective patients from retrospective patients. Baseline characteristics were similar between the subgroups, supporting the overall study findings and reducing the chance of reporting bias presence.

To conclude, fingolimod treatment sequence after natalizumab was found to be safe with no new safety concerns during a two-year observation after switching from natalizumab. The incidence of AEs decreased with increase in treatment duration. Fingolimod was effective in controlling the disease activity over two years. Our study demonstrates that the benefit-risk profile supports a washout period of less than eight weeks in patients transitioning from natalizumab to fingolimod.

## Figures and Tables

**Figure 1 brainsci-12-00215-f001:**
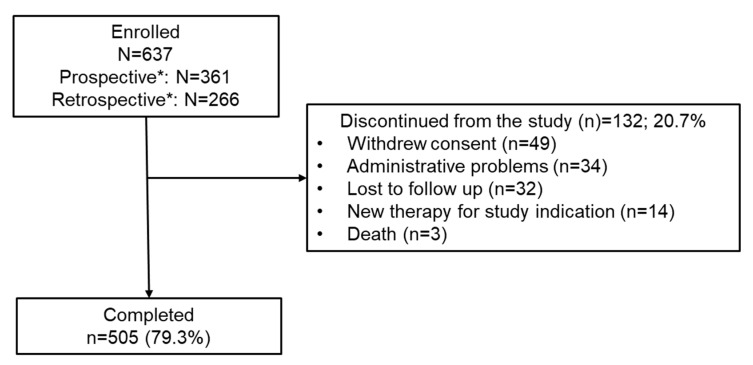
Patient disposition. * post hoc analysis; 10 patients could not be categorized into prospective/retrospective groups.

**Figure 2 brainsci-12-00215-f002:**
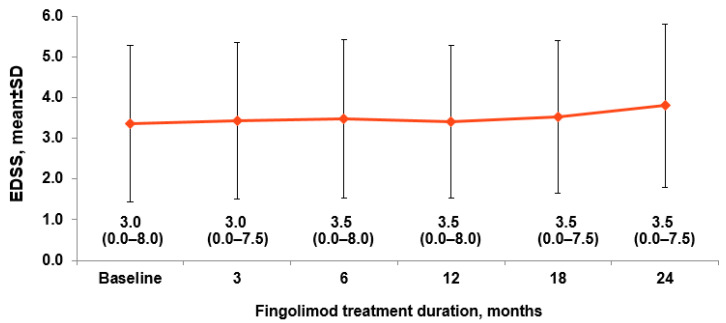
EDSS score over 24 months, by visit. Data mentioned below the error bars are medians (ranges); analysis included safety set EDSS, Expanded Disability Status Scale; SD, standard deviation.

**Figure 3 brainsci-12-00215-f003:**
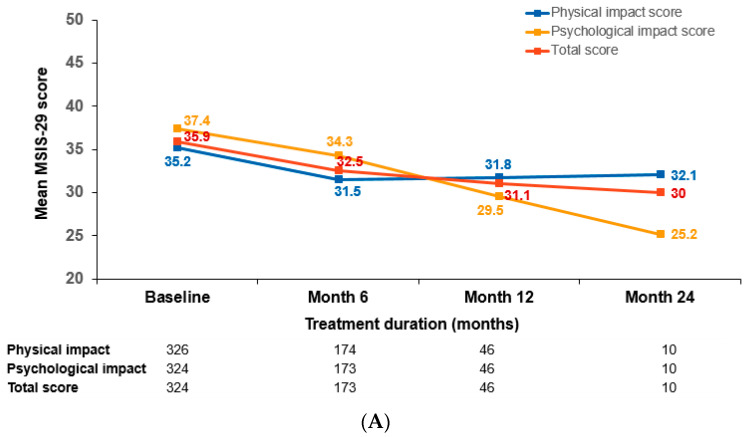
Mean MSIS-29 (**A**) and TSQM-9 (**B**) scores by time. MSIS, Multiple Sclerosis Impact Scale; TSQM-9, Treatment Satisfaction Questionnaire for Medication-9. Analysis included safety set.

**Table 1 brainsci-12-00215-t001:** Patient demographics and baseline characteristics.

Characteristics	Fingolimod 0.5 mg N = 637	Prospective N = 361 ^a^	Retrospective N = 266 ^a^
**Age, years**	42.0 *±* 10.4	42.4 *±* 10.6	41.4 *±* 10.1
**Women, n (%)**	458 (71.9)	267 (74.0)	185 (69.5)
**Caucasian, n (%)**	580 (91.1)	335 (92.8)	240 (90.2)
**MS duration since diagnosis, years**	10.5 *±* 6.8	10.4 *±* 6.9	10.6 *±* 6.7
**Number of relapses in the last 12 months ^b^**	0.5 *±* 0.8	0.47 ± 0.76	0.50 *±* 0.87
**EDSS score**	3.4 *±* 1.9	3.43 *±* 1.94	3.25 *±* 1.88
**JCV status known, n (%)**	613 (96.2)	354 (98.1)	253 (95.1)
**JCV-positive patients, n (%)**	512 (83.5)	288 (81.4)	218 (86.2)
	**Fingolimod 0.5 mg** **N = 628**		
**Any prior MS DMTs at study entry, n (%) * ^**	626 (99.7)	361 (100)	265 (99.6)
**Most common prior MS DMTs**			
**Natalizumab**	625 (99.5)	360 (99.7)	265 (99.6)
**Interferon β-1a**	334 (53.2)	180 (49.9)	154 (57.9)
**Glatiramer acetate**	226 (36.0)	138 (38.2)	88 (33.1)
**Interferon β-1b**	90 (14.3)	45 (12.5)	45 (16.9)
**Betaseron**	58 (9.2)	42 (11.6)	16 (6.0)
**Washout period**			
**Duration, weeks**	14.5 *±* 10.9	14.9 *±* 11.3	13.9 *±* 10.4
**≤8 weeks, n (%)**	156 (24.8)	87 (24.1)	69 (25.9)
**>8 weeks, n (%)**	470 (74.8)	273 (75.6)	197 (74.1)

Data from enrolled set are presented as mean ± standard deviation, unless stated otherwise. ^a^ 10 patients could not be categorized into prospective/retrospective groups. ^b^ includes relapses during washout period. * Percentages are calculated using the total number of patients (N = 628) in the safety set as the denominator. ^ Defined as MS DMTs that started and ended on or before the first dose of fingolimod. DMTs, disease-modifying therapies; EDSS, Expanded Disability Status Scale; JCV, John Cunningham virus.

**Table 2 brainsci-12-00215-t002:** Incidences of AEs (>4% of patients in all patients) and SAEs (>2% of patients).

Events	All Patients N = 628		Prospective Patients (N = 361)		Retrospective Patients (N = 266)	
	n (%)	IR (95% CI)	n (%)	IR (95% CI)	n (%)	IR (95% CI)
**Total AEs**	**457 (72.8)**	**91.49** **(83.29; 100.27)**	**272** **(75.4)**	**104.13** **(92.12; 117.26)**	**185 (69.6)**	**77.6** **(66.85; 89.66)**
Fatigue	74 (11.8)	6.4 (5.04; 8.06)	45 (12.5)	6.9 (5.01; 9.19)	29 (10.9)	5.8 (3.90; 8.37)
Headache	61 (9.7)	5.2 (3.97; 6.67)	38 (10.5)	5.7 (4.04; 7.83)	23 (8.7)	4.5 (2.86; 6.78)
Urinary tract infection	57 (9.1)	4.8 (3.64; 6.22)	39 (10.8)	5.9 (4.16; 7.99)	18 (6.8)	3.5 (2.05; 5.47)
Lymphopenia	44 (7.0)	3.7 (2.66; 4.91)	28 (7.8)	4.1 (2.73; 5.94)	16 (6.0)	3.1 (1.76; 4.99)
Depression	41 (6.5)	3.4 (2.43; 4.59)	26 (7.2)	3.8 (2.47; 5.54)	15 (5.6)	2.9 (1.60; 4.71)
Diarrhea	33 (5.3)	2.7 (1.87; 3.81)	21 (5.8)	3.0 (1.88; 4.64)	12 (4.5)	2.3 (1.18; 3.99)
Muscular weakness	32 (5.1)	2.6 (1.79; 3.69)	19 (5.3)	2.7 (1.64; 4.25)	13 (4.9)	2.5 (1.32; 4.23)
Constipation	31 (4.9)	2.5 (1.71; 3.58)	22 (6.1)	3.2 (1.99; 4.80)	9 (3.4)	1.7 (0.77; 3.20)
Back pain	30 (4.8)	2.5 (1.66; 3.50)	16 (4.4)	2.3 (1.32; 3.74)	14 (5.3)	2.7 (1.45; 4.46)
Fall	28 (4.5)	2.3 (1.51; 3.29)	15 (4.2)	2.1 (1.20; 3.53)	13 (4.9)	2.5 (1.31; 4.19)
Insomnia	27 (4.3)	2.2 (1.45; 3.19)	14 (3.9)	2.0 (1.09; 3.35)	13 (4.9)	2.5 (1.31; 4.21)
Hypertension	26 (4.1)	2.1 (1.39; 3.11)	13 (3.6)	1.9 (0.99; 3.17)	13 (4.9)	2.5 (1.33; 4.26)
Nausea	26 (4.1)	2.1 (1.39; 3.11)	20 (5.5)	2.9 (1.77; 4.48)	6 (2.3)	1.1 (0.41; 2.44)
**Total SAEs**	**80 (12.7)**	**6.9 (5.50; 8.63)**	**47 (13.0)**	**7.2** **(5.30; 9.59)**	**33 (12.4)**	**6.6 (4.53; 9.24)**
Multiple sclerosis relapse	13 (2.1)	1.0 (0.56; 1.78)	7 (1.94)	1.0 (0.40; 2.04)	6 (2.3)	1.1 (0.41; 2.42)

IR is defined as the number of patients who reported at least one AE in this category, over the total patient-years of the population for that event. Patients were censored at the time of the event. IR is expressed per 100 patient-years of the population; analysis included safety set. AEs, adverse events; CI, confidence interval; IR, incidence rate; n, number of patients; PT, preferred term; SAEs, serious adverse events.

**Table 3 brainsci-12-00215-t003:** Change in MRI outcomes.

	Pre-Baseline			Month 12			Month 24		
	≤8 Weeks	>8 Weeks	Overall Population	≤8 Weeks	>8 Weeks	Overall Population	≤8 Weeks	>8 Weeks	Overall Population
MRI performed, n	154	449	605	46	159	205	15	39	54
Worsening in MRI since the last scan was performed, n (%)	24 (15.6)	86 (19.2)	110 (18.2)	11 (23.9)	31 (19.5)	42 (20.5)	4 (26.7)	10 (25.6)	14 (25.9)
Presence of Gd+ T1 lesions	8	39	47	5	7	12	1	4	5
Increase in number of T2 lesions	15	42	57	5	16	21	4	5	9
Other	6	17	23	3	12	15	0	2	2

Gd+, gadolinium-enhancing. MRI was not mandatory per protocol. Data collected from patients, where available.

## Data Availability

The data presented in this study are available on request from the corresponding author. These data were used under license for the current study, and so are not publicly available.

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
