# Peer review of "Safety of Fingolimod in Patients with Multiple Sclerosis Switched from Natalizumab: Results from TRANSITION―A 2-Year, Multicenter, Observational, Cohort Study"

_brainsci, 2022, doi:10.3390/brainsci12020215_

Round 1

Reviewer 1 Report

1) Methods. Please provide the IRB number and what was the ethical committee location.

2) About questionnaires:

a) Please provide if authorization for use of MSIS and TSQM were requested and write this in the methods.

b) Were there no other questions asked besides those related to these two questionnaires?

3) How was the data distribution about normality?

4) Please provide a ‘‘Figure S1. TRANSITION study design’’ of high-quality.

5) ‘‘2.2 Patient population’’ Why was 12 months chosen as the period? Provide evidence for this chosen period.

6) In the limitations of the study, add information about the significant discontinuation of the study.

7) Figure 1 and Table 1. Missing information

Enrolled N=637 Prospective 361 Retrospective 266

361 + 266 = 627

8) Were anatomical location and quantity of MS lesions studied? This should be addressed with a specific multivariate analysis to exclude this confounding variable. Or if this does not change the results, the authors should describe in the discussion section this drawback and provide evidence that this data does not influence the final results.

9) Data is similar to the TOFINGO study. The authors should describe what does this manuscript brings new to the present literature?

Kappos L, Radue EW, Comi G, Montalban X, Butzkueven H, Wiendl H, Giovannoni G, Hartung HP, Derfuss T, Naegelin Y, Sprenger T, Mueller-Lenke N, Griffiths S, von Rosenstiel P, Gottschalk R, Zhang Y, Dahlke F, Tomic D; TOFINGO study group. Switching from natalizumab to fingolimod: A randomized, placebo-controlled study in RRMS. Neurology. 2015 Jul 7;85(1):29-39. DOI: 10.1212/WNL.0000000000001706. Epub 2015 May 29. PMID: 26024899; PMCID: PMC4501941.

Jokubaitis VG, Li V, Kalincik T, Izquierdo G, Hodgkinson S, Alroughani R, Lechner-Scott J, Lugaresi A, Duquette P, Girard M, Barnett M. Fingolimod after natalizumab and the risk of short-term relapse. Neurology. 2014 Apr 8;82(14):1204-11.

10) It is advised for the authors to provide a section about financial support and conflicts of interest. Mainly if received any assistance from the pharmaceutical industry.

11) The title should mention that is a ‘‘cohort’’ and that is ‘‘multicenter’’.

NEW IDEA: provide a supplemental table with the information of other studies already published in the literature comparing with the present like year, number of patient, adverse events, and other.

Reviewer 2 Report

The authors addressed a timely topic in the treatment of multiple sclerosis. Most of the paper is devoted to the results of the analysis of the incidence of adverse events occurring after conversion of natalizumab treatment to fingolimod treatment. The authors emphasise in their paper that the main objective was to determine the safety for the patient of such a conversion. This aspect seems to have been quite well studied. An additional aim was to present the effectiveness of fingolimod treatment, which is emphasised by the title of the paper. In my opinion, however, this aspect of the paper was not well developed. If treatment efficacy is already emphasised in the title as one of the objectives, the study should have been designed differently. I believe that performing MRI scans throughout the follow-up period should have been a mandatory inclusion criterion for the study population. I am also unsatisfied with the way the treatment efficacy criteria were presented. EDSS progression and MRI changes were presented superficially, the statistical method could have included survival analysis (Kaplan-Meyer). In light of the available literature, a separate discussion of the three parameters of disease progression is not a method compliant with current standards, e.g. measures such as Rio or modified Rio were missing. The vast majority of results were presented in numerical form, which does not facilitate their perception. There is a lack of thoughtful graphic illustrations. I also believe that the literature selection is not adequate - there is a lack of literature items from 2019-2021, several cited are from 15 years ago. The literature is too sparse - it has only 26 items that do not relate to recent research. 

Reviewer 3 Report

The study presented offers a relevant real-life data on the shift to fingolimod after natalizumab in RRMS. The risk of PML and outcome is discussed with 24-months follow up.

This is indeed a high profile manuscript offering relevant insight for a wide audience of clinicians. 

I suggest to clarify the sites of enrollment of the subjects and enrollment period (start and stop date) . There is an overall variability in the diagnosis of PML cases due to national bias in the follow up of MS patients and MRI access to unified screening. In various times was also suggested to establish a specific registry. Also the diagnostic tests during time changed in specificity and sensitivity.

I suggest to add in the method section or as supplement material a short description of the recruitment site involved over time or at least the list of involved nations with relative numerosity.

e.g. Establishment of a Registry To Identify Cases of PML Using Uniform Diagnostic Criteria (S08.002) Avindra Nath, Eugene Major, Allen Aksamit, Leonard Calabrese, James Sejvar, Camille Kotton, Maria Barhams

Round 2

Reviewer 2 Report

Thank you for your reply and changes that addressed my previous comment. At it's present form, I believe the paper is ready for publication.